# Overexpression of ITGB3 in Peripheral Blood Mononuclear Cells of Relapsing-Remitting Multiple Sclerosis Patients

**DOI:** 10.3390/ijms262412094

**Published:** 2025-12-16

**Authors:** Giselle Berenice Vela Sancho, Ricardo E. Buendia-Corona, María Paulina Reyes-Mata, Mario Alberto Mireles-Ramírez, Christian Griñán-Ferré, Mercè Pallàs, Ana Laura Márquez-Aguirre, Lenin Pavon, Oscar Arias-Carrión, José de Jesús Guerrero-García, Daniel Ortuño-Sahagún

**Affiliations:** 1Laboratorio de Neuroinmunobiología Molecular, Instituto de Neurociencias Translacionales, Centro Universitario de Ciencias de la Salud (CUCS), Universidad de Guadalajara, Guadalajara 44340, Mexico; giselle.vela3080@alumnos.udg.mx; 2Department of Chemical Biological Sciences, Universidad de las Américas Puebla, Puebla 72810, Mexico; ricardo.buendiaca@udlap.mx; 3Departamento de Disciplinas Filosófico, Metodológicas e Instrumentales, Centro Universitario de Ciencias de la Salud (CUCS), Universidad de Guadalajara, Guadalajara 44340, Mexico; paulina.reyes@academicos.udg.mx; 4Unidad Médica de Alta Especialidad (UMAE), Hospital de Especialidades (HE), Centro Médico Nacional de Occidente (CMNO), Instituto Mexicano del Seguro Social (IMSS), Guadalajara 44340, Mexico; drmmireles@gmail.com; 5Pharmacology Section, Department of Pharmacology, Toxicology and Therapeutic Chemistry, Faculty of Pharmacy and Food Sciences, Institute of Neuroscience, Universitat de Barcelona, 08028 Barcelona, Spain; christian.grinan@ub.edu (C.G.-F.); pallas@ub.edu (M.P.); 6CiberNed, Network Center for Neurodegenerative Diseases, National Spanish Health Institute Carlos III, 28220 Madrid, Spain; 7Unidad de Biotecnología Médica y Farmacéutica, Centro de Investigación y Asistencia en Tecnología y Diseño del Estado de Jalisco A.C. (CIATEJ), Guadalajara 44270, Mexico; amarquez@ciatej.mx; 8Laboratorio de Psicoinmunología, Instituto Nacional de Psiquiatría Ramón de la Fuente Muñiz, Mexico City 14370, Mexico; lkuriaki@inprf.gob.mx; 9División de Neurociencias, Clínica, Instituto Nacional de Rehabilitación Luis Guillermo Ibarra Ibarra, Mexico City 14389, Mexico; ariasemc2@gmail.com; 10Escuela de Medicina y Ciencias de la Salud, Tecnologico de Monterrey, Mexico City 14380, Mexico; 11Banco Central de Sangre, Unidad Médica de Alta Especialidad (UMAE), Hospital de Especialidades (HE), Centro Médico Nacional de Occidente (CMNO), Instituto Mexicano del Seguro Social (IMSS), Guadalajara 44340, Mexico; 12Departamento de Farmacobiología, Centro Universitario de Ciencias Exactas e Ingenierías (CUCEI), Universidad de Guadalajara, Guadalajara 44340, Mexico

**Keywords:** pleiotrophin, multiple sclerosis, integrin receptor, ITGB3, molecular docking

## Abstract

Multiple sclerosis (MS), the most prevalent chronic inflammatory, demyelinating and neurodegenerative disease of the central nervous system in young adults, exhibits marked sexual dimorphism, with a 3:1 female-to-male ratio, but more severe symptoms and greater neurological damage in males. Increasing attention has focused on identifying circulating molecules that reflect inflammatory activity within the central nervous system and could clarify the mechanisms underlying MS. Pleiotrophin (PTN), a cytokine implicated in autoimmune and neurological diseases, is significantly elevated in patients with relapsing-remitting MS (RRMS). To explore the potential contribution of PTN and its receptors to neuroinflammatory signaling, we quantified the mRNA expression of PTN receptors in peripheral blood mononuclear cells from RRMS patients compared to untreated RRMS patients and healthy control subjects. We further performed an in silico molecular docking and molecular dynamics analysis to assess the possible functional significance of PTN-receptor interactions. Our results show a significant overexpression of *integrin subunit beta-3* (*ITGB3*) mRNA in peripheral blood mononuclear cells from RRMS patients compared to healthy control subjects. Molecular docking shows that PTN could binds to the metal ion-dependent adhesion site domain of ITGB3 via Mg^2+^/Ca^2+^-mediated stabilization and has a higher binding affinity than fibrinogen, the canonical endogenous ligand. These findings suggest that ITGB3 could be a dynamically regulated integrin receptor in RRMS that may participate in PTN-driven neuroinflammatory pathways in peripheral blood immune cells, influenced by disease stage, sex, and immunotherapy. While our results support the biological plausibility of PTN–ITGB3 engagement, they remain hypothesis-generating and require functional validation. The integration of molecular expression data and computational modeling underscores the potential involvement of ITGB3 as a possible participant in MS and warrants further investigation of its clinical and mechanistic role.

## 1. Introduction

Multiple sclerosis (MS) is the most prevalent chronic inflammatory, demyelinating, and neurodegenerative disease of the central nervous system (CNS) affecting young adults [1,2]. As a leading cause of non-traumatic neurological disability in this population [3], MS is primarily an immune-mediated disease characterized by clinical heterogeneity and manifests in diverse forms, such as relapsing-remitting MS (RRMS), which accounts for 80–85% of cases. RRMS is defined by episodes of clinical activity (relapses) followed by periods of complete or partial remission [4,5].

Notably, MS exhibits pronounced sexual dimorphism, with a 3:1 female-to-male ratio, although males often have more severe symptoms and greater neurological damage [6]. Despite extensive research MS remains incurable. Current treatment relies on disease-modifying therapies (DMTs), which aim to modulate immune responses, reduce the frequency of relapses and slow the progression of brain lesions to limit neurological damage [7].

First-generation DMTs, such as interferon beta (IFN-β) and glatiramer acetate (GA), show variable efficacy, with many patients being refractory or suffering from adverse side effects. Second-generation DMTs, including natalizumab (NAT), rituximab (RIT) and fingolimod (FINGO) [8], offer higher efficacy but are associated with higher toxicity [9,10]. The clinical heterogeneity of MS and the variable response to treatment underscore the urgent need for personalized therapeutic strategies based on a deeper understanding of disease mechanisms.

Recent research has focused on the identifying of molecular changes in peripheral blood that reflect inflammatory processes in the CNS and provide insights into the pathophysiology of MS [11,12,13]. Elevated serum levels of pleiotrophin (PTN), a neuroimmunomodulatory cytokine implicated in autoimmune and neurological diseases, have been observed in RRMS patients compared to healthy control subjects (HCS) [14]. Building on our previous findings that demonstrated PTN overexpression in RRMS patients relative to HCS [14], the present study aims to investigate the PTN signaling axis in greater mechanistic detail by examining the expression of its principal receptors (PTPRZ, ALK, SDC3, and ITGB3) in PBMCs. PTN was selected as a candidate molecule based on prior evidence of its relevance in neuroinflammatory and remyelination processes, and on the emerging role of microRNAs regulating PTN expression in MS [15]. Additionally, specific microRNAs that regulate PTN expression at the post-transcriptional level have been associated with MS [15], suggesting a complex regulatory network involving PTN in the pathogenesis of the disease.

Although PTN interacts with integrin β3 (ITGB3), a receptor required for PTN-induced endothelial migration through modulation of receptor protein tyrosine phosphatase ζ (RPTPζ), linking PTN signaling with cellular adhesion, migration, and angiogenesis [16,17,18]. Given that integrins are highly expressed in circulating immune cells, including peripheral blood mononuclear cells (PBMCs), these interactions suggest that PTN receptors may also influence peripheral immune cell function. In MS, peripheral immune cells play a critical role in neuroinflammation and contribute to CNS pathology through infiltration across the brain blood barrier (BBB). Thus, analyzing PTN receptor expression in PBMCs offers an opportunity to connect peripheral immune dysregulation with CNS mechanisms, providing insight into their contribution to neurodegeneration [19].

Given the role of PTN and its receptors in neuroinflammatory processes, including oligodendrocyte differentiation and remyelination [20], this study investigates the expression profiles of PTN receptors in PBMCs from RRMS patients compared to untreated patients and HCS. We also investigate the correlations between receptor expression levels and clinical variables. By elucidating the role of PTN receptors in MS, we aim to uncover novel pathophysiological pathways and contribute to the development of precise diagnostic and therapeutic strategies for MS patients. Accordingly, the present study focuses on transcriptional profiling and in silico structural analyses to prioritize ITGB3 within the PTN axis. We do not test receptor-mediated signaling experimentally; rather, we aim to generate mechanistic hypotheses to guide subsequent validation.

## 2. Results

Demographic and clinical characteristics of the study population are summarized in Table 1. The RRMS group had a balanced sex distribution, a mean EDSS score consistent with low disability, and no significant differences in age or body mass index (BMI) compared to HCS.

### 2.1. Expression Analysis of PTN Receptors in PBMC from RRMS Patients and HCS

To elucidate the role of PTN receptors in RRMS, we first analyzed PBMCs from a subgroup of RRMS patients treated with IFN-β (n = 33) and compared them with HCS (n = 42) using RT-qPCR. In this initial screen, none of the tested PTN receptors—RPTPζ, anaplastic lymphoma kinase (ALK), ITGB3, and syndecan-3 (SDC3)—showed detectable expression in the IFN-β subgroup. This suggests a treatment-related suppression consistent with the known immunomodulatory effects of IFN-β, which downregulates integrin expression and reduces leukocyte adhesion and migration. Because no expression was observed in IFN-β–treated patients, we next expanded the analysis to the broader RRMS cohort, which included patients under different DMTs (RIT, NAT, FINGO) and treatment naïve individuals (those who have not received any DMT), as detailed in Table 1. In this extended analysis, only *ITGB3* exhibited consistent mRNA expression in both RRMS patients and HCS, while PTPRZ1, ALK, and SDC3 remained undetectable in all samples. Consequently, subsequent analyses focused exclusively on *ITGB3*, which emerged as the main PTN receptor expressed in PBMCs of RRMS patients. The specificity and reliability of the RT-qPCR assays were validated through electropherograms (Appendix A), confirming accurate amplification of the targeted PTN receptors. These results underscore the potential relevance of ITGB3 as a possible target of interest in RRMS and warrant further investigation into its role in disease pathogenesis and therapeutic response.

### 2.2. Relative Expression of ITGB3 in RRMS Patients Compared to HCS

To investigate *ITGB3* gene expression in RRMS patients, RT-qPCR analysis was performed using 18S as an endogenous control. As shown in Figure 1A, *ITGB3* expression was significantly higher in RRMS patients compared to HCS (*p* < 0.05). Figure 1B shows lower ΔCt values in the RRMS group, confirming increased expression levels. These differences were statistically significant (*p* < 0.05).

### 2.3. Sex-Specific Differences in ITGB3 Expression

In addition, we observed sex-specific differences in *ITGB3* expression (Figure 2A). Male RRMS patients had the highest *ITGB3* expression levels, followed by female RRMS patients, while both male and female HSCs showed lower expression. Figure 2B confirms these results using ΔCt values: male RRMS patients showed significantly higher expression compared to male HSCs (*p* < 0.05), whereas the difference between female RRMS patients and female HSCs was not statistically significant.

### 2.4. ITGB3 Expression in Naïve and DMT-Treated RRMS Patients

We found that naïve RRMS patients exhibited significantly higher *ITGB3* expression compared to HCS (Figure 3B,D; *p* < 0.05). These findings suggest that *ITGB3* overexpression is present early in the disease course, even before therapy initiation.

When *ITGB3* expression was compared across DMTs, all RRMS subgroups showed higher expression than HCS, with the highest levels observed in naïve and RIT-treated patients (Figure 3C). However, statistical analysis of ΔCt values revealed a significant difference only between HCS and the naïve group (Figure 3D). This further supports that *ITGB3* overexpression is most prominent in untreated RRMS patients.

### 2.5. Correlation Between ITGB3 Expression, Time Since Diagnosis, and Disease Duration

In the overall treated RRMS cohort, *ITGB3* expression did not show a clear correlation with either year since diagnosis or disease duration. Patients with longer histories tended to have slightly higher expression, but these changes were inconsistent. Based on visual inspection of ΔCt data (Figure 4A,C,E,G), a noticeable shift in *ITGB3* expression patterns was observed around 10 years of disease duration. This empirical separation used a 10-year cutoff to stratify patients, revealing significant differences in *ITGB3* expression between early (<10 years) and long-standing (>10 years) RRMS (Figure 4F,H).

A different pattern emerged in patients treated with RIT or NAT. In this subgroup, *ITGB3* expression increased with longer time since diagnosis, suggesting that prolonged disease history under these therapies is associated with higher expression. In contrast, when considering disease duration, the trend was reversed: patients with longer disease duration showed a decline in *ITGB3* expression. These results indicate a complex relationship between *ITGB3* expression, disease chronicity, and therapeutic context, with opposing trends depending on whether time since diagnostic or disease duration is considered.

### 2.6. Correlation Between ITGB3 Expression and Time Since Diagnosis

Correlation analysis indicated that *ITGB3* expression tended to increase with years of diagnosis in RRMS patients under treatment, although the overall association was not statistically significant (Figure 4A). When patients were stratified by treatment, those receiving RIT, FINGO, or NAT with more than 10 years since diagnosis exhibited significantly higher expression compared to patients with fewer years of diagnosis (Figure 4B). Notably, in the subgroup of patients treated with RIT and NAT, *ITGB3* expression correlated strongly and positively with time since diagnosis (Figure 4E,F). These findings suggest that *ITGB3* expression may rise with disease chronicity, particularly in RRMS patients treated with monoclonal antibody therapies.

### 2.7. Correlation Between ITGB3 Expression and Disease Duration

Analysis of the overall RRMS cohort showed no significant correlation between *ITGB3* expression and disease duration (Figure 4C,D). However, when the analysis was limited to patients receiving RIT or NAT, a different pattern appeared. In this subgroup, *ITGB3* expression correlated positively with time since diagnosis (Figure 4G) and was significantly lower in patients with more than 10 years of disease duration compared to those with a shorter history (Figure 4H). These results suggest that *ITGB3* expression may decline over time in RRMS patients treated with monoclonal antibody therapies, reflecting potential treatment-associated changes during disease progression.

### 2.8. Molecular Docking PTN-ITGB3

Molecular docking shows that PTN123-132 binds to the B3 region of ITGB3 with a predicted binding energy of −5.6 kcal/mol, compared to −5.1 kcal/mol for fibrinogen (Figure 5). However, with three independent docking runs per ligand (9 modes each), the average predicted binding affinities were −4.53 ± 0.40 kcal/mol for PTN and −4.43 ± 0.56 kcal/mol for fibrinogen (Appendix A). GLN131 (numbering according to PDB ID: 2N6F) forms an unfavorable donor/donor interaction, while Lys129 and Lys130 establish coordination-like contacts with divalent cations (Mg^2+^ and Ca^2+^) at the MIDAS site. These metal ion interactions contribute to favorable binding between PTN and the integrin receptor. The diverse biological functions of PTN result from the interactions at the N- and C-terminal regions with its receptors, which lead to various functions [22]. Understanding the molecular interaction of PTN with ITGB3 opens the door to further exploration of its biological function in MS.

### 2.9. Molecular Dynamics of the PTN–ITGB3 Complex

To evaluate the dynamic stability and interaction profile of the PTN–ITGB3 complex, three MD models were simulated over 250 ns. Below, a summary of the main structural and energetic parameters is presented, including RMSD, interatomic contacts, hydrogen bonding, and binding free energy. Detailed results for each parameter are discussed in the following subsections.

These simulations provided crucial insights into the maintenance and intermolecular interactions of the complex. In particular, the RMSD was analyzed both for the complete ITGB3 receptor and for a version excluding the first 59 residues, corresponding to a disordered N-terminal loop. The RMSD plot (Figure 6) shows values for PTN (magenta), ITGB3 receptor (orange), and receptor without the flexible N-terminal region (residues 1–59) (olive green), which shows less variability (Table 2). This suggests that the high fluctuations observed in the canonical RMSD are largely due to the flexibility of the N-terminal loop and not to the global instability of the complex.

As for the interatomic contacts (Figure 6), both the native and non-native ITGB3–PTN complexes retained a considerable number of interactions throughout the simulation (Table 2). The non-native complex (shown in dark orange) exhibited a higher average number of contacts than the native structure (light orange). Despite the greater number of contacts, the non-native complex also exhibited greater atomic fluctuations, as reflected by higher RMSD values in key binding regions. This indicates less structural retention compared to the native complex, where contact formation appeared more consistent and structurally restrained.

When analyzing the hydrogen bonds (Figure 6), the contributions of the donor atoms of the receptor (cyan) and the ligand PTN (pink) were quantified. Of note, PTN consistently formed more hydrogen bonds than ITGB3 in Figure 6 (Table 2), suggesting a greater role in steadying the interaction network. Although the other values were more similar between ligand and receptor, the overall pattern highlights the significant contribution of PTN in preserving the complex through specific, directional interactions.

Finally, the free energies of binding calculated by the MM-GBSA method (Figure 6) indicate a favorable interaction between PTN and ITGB3, as shown by the consistently negative ΔG values. Although MM-GBSA provides an approximate estimate, the persistence of negative values during the simulation suggests sustained and energetically favorable binding (Table 2). Importantly, this favorable profile was maintained despite variations in the number of interatomic contacts and hydrogen bonds, suggesting that key interactions at the binding interface are both strong and partially redundant. Overall, the model supports a durable association between PTN and ITGB3 and emphasizes the biological plausibility of the predicted complex.

Finally, the binding free energy calculated by the MM-GBSA method (Figure 6) for the native complex shows a favorable interaction, as evidenced by negative ΔG values (Table 2). This indicates that the PTN–ITGB3 binding process is thermodynamically spontaneous. Throughout the MD simulation, the binding energy remains relatively consistent, despite natural fluctuations in contact numbers and hydrogen bonds. This persistence suggests that key interactions at the binding site are strong and redundant, supporting the endurance of the complex. The overall energetic profile over time emphasizes the robustness of the interaction and reinforces the hypothesis that the PTN–ITGB3 complex is potentially biologically relevant.

### 2.10. Bioinformatic Analysis of Predicted and Validated Possible Amino Acids for PTN-ITGB3 Interaction by Molecular Dynamics Study and Multiple Sequence Alignment

To further investigate the molecular basis of PTN binding to the ITGB3 receptor, we performed MD and subsequent MD simulations. Docking analysis revealed that the C-terminal peptide of PTN123-132 binds to the B3 region of ITGB3 with a binding energy of −5.6 kcal/mol, slightly stronger than the known ligand fibrinogen (−5.1 kcal/mol). The most important interactions include Lys129 and Lys130, which coordinate with Ca^2+^ and form hydrogen bonds, while Gln131 forms both favorable and unfavorable interactions with Mg^2+^ at the MIDAS motif. These contacts mediated by the metal ions appear to stabilize the PTN–ITGB3 complex. To evaluate the structural and durability effects of this interaction, we performed a 250 ns MD simulation. The results highlight key conserved residues involved in binding and show consistent interactions throughout the simulation.

To identify conserved amino acids in different species, an MSA was performed with consensus sequences from different organisms in Jalview 2.11.4.0. This approach allowed the visualization of conserved regions within the ITGB3 sequence and highlighted residues that could play a crucial role in structural or functional integrity. These conserved residues were then compared to those involved in the ITGB3–PTN interactions observed during MD (Figure 7). This provided insight into the evolutionary significance of the binding interface and confirmed the relevance of the identified interaction sites.

## 3. Discussion

The present study identifies ITGB3 as the only PTN receptor consistently expressed and overexpressed in PBMCs from patients with RRMS. This finding is disruptive for two reasons. First, it extends the biology of PTN beyond the CNS, where it has traditionally been studied, and positions ITGB3 as a peripheral immune receptor relevant to neuroinflammation. Second, it reframes integrin signaling in MS: while α4 integrins have long been established as central to lymphocyte trafficking across the BBB [23] and serve as therapeutic targets for NAT, our findings suggest that ITGB3 could represent an alternative or complementary immune checkpoint integrin modulating peripheral immune activity, BBB dynamics, and disease progression.

Our results demonstrate a ~3.1-fold increase in *ITGB3* expression in PBMCs from RRMS patients compared to HCS, indicating that ITGB3 is not merely a bystander receptor but may play an active role in the immunopathology of MS. The magnitude of this overexpression was most pronounced in treatment-naïve patients, suggesting that *ITGB3* dysregulation occurs early in the disease process, prior to therapeutic intervention. Additionally, male patients exhibited the highest levels of *ITGB3*, consistent with the clinical paradox of MS: while women are more frequently affected, men tend to experience more severe disability and faster progression [6,24,25]. This sex-specific signature underscores the need to dissect how integrin-mediated signaling is differentially regulated across biological contexts and may inform sex-sensitive therapeutic strategies.

Treatment context also influenced *ITGB3* expression. Patients receiving RIT or NAT showed sustained or even increased *ITGB3* expression with longer disease duration, whereas IFN-β treatment was associated with undetectable receptor expression. This likely reflects the transcriptional downregulation of adhesion molecules induced by IFN-β, in contrast to other DMTs, such as NAT or FINGO, which act through distinct mechanisms and do not directly suppress gene expression. These results indicate that *ITGB3* regulation is dynamic and reflects both disease chronicity and immunotherapy [26]. This pattern raises the provocative possibility that ITGB3 acts as an adaptive integrin checkpoint, modulating immune–vascular interactions in the face of prolonged disease and pharmacological intervention. Given that αvβ3 integrins regulate leukocyte adhesion and endothelial permeability [16,27,28]. *ITGB3* overexpression may contribute to the remodeling of immune migration across the BBB [26]. In this context, ITGB3 could represent a new axis of therapeutic interest, potentially complementing or counteracting current integrin-targeting therapies in MS.

Our structural analyses provide an additional layer of evidence by integrating patient-derived transcriptomic data with molecular modeling. Although the PTN–ITGB3 interaction has been previously described, the objective of our analysis was to characterize the energetic and structural stability of the complex within the context of MS-related molecular alterations. By coupling molecular modeling with differential gene expression data, these findings provide an integrated perspective linking *ITGB3* upregulation to PTN-mediated signaling. Docking studies revealed PTN binding at the MIDAS site with predicted affinities comparable to fibrinogen, the canonical endogenous ligand [29], thereby supporting the biological relevance of the PTN–ITGB3 interaction in disease-related settings. This observation reinforces the plausibility that PTN–ITGB3 signaling is not incidental but represents a biologically relevant pathway. Although limited to 250 ns and conducted without membrane bilayers, MD simulations revealed persistent binding energies and sustained hydrogen bond networks, indicating a stable and energetically favorable PTN–ITGB3 complex. These computational results should not be interpreted as definitive proof of interaction but rather as hypothesis-generating insights, placing PTN–ITGB3 engagement within a structural framework that requires in vitro validation. Future studies using patient-derived immune cells stimulated with PTN, or BBB-on-chip models that recapitulate vascular complexity, will be essential to determine the functional consequences of this interaction.

The biological plausibility of ITGB3 involvement in MS is further supported by its established role in endothelial cell migration, angiogenesis, and immune cell adhesion [16,30,31]. Integrin αvβ3 interacts with fibrinogen and osteopontin, both elevated in MS lesions [32,33], suggesting convergent pathways that may amplify neuroinflammation. Moreover, ITGB3 is involved in T cell polarization, with higher expression reported in myelin basic protein–primed Th2 cells compared to Th1 subsets [28]. This raises the intriguing possibility that ITGB3 may regulate not only immune cell trafficking but also the balance of pro- and anti-inflammatory T cell responses in MS. By connecting these findings, our results suggest that PTN–ITGB3 signaling could function as a molecular switch, shaping the immunological landscape of RRMS through effects on adhesion, migration, and effector polarization [34].

This study was conceived as an exploratory screening aimed at characterizing the gene expression profile of PTN receptors and providing structural validation through complementary bioinformatic analyses. Accordingly, *ITGB3* expression was evaluated at the mRNA level, molecular docking and molecular dynamics simulations were incorporated to strengthen the biological interpretation of the transcriptional data. Although the interaction between PTN and ITGB3 has not previously been implicated in MS, our molecular dynamics simulations show that their binding is stabilized by multiple intermolecular forces. Because PTN levels are elevated in patients with RRMS and correlate with sex and BMI [14], the stability of the PTN–ITGB3 complex may enhance integrin-mediated signaling through the SRC and PI3K–AKT pathways [20,35]. These pathways promote immune cell migration and could facilitate leukocyte infiltration into the CNS across the BBB. This previously unrecognized PTN–ITGB3 axis could represent a novel mechanistic node in MS pathogenesis that warrants further functional validation and exploration as a potential therapeutic target to modulate CNS immune cell trafficking and disease heterogeneity.

While functional and protein-level validation were beyond the scope of this work, our integrated approach establishes a solid basis for subsequent studies investigating PTN–ITGB3 signaling mechanisms in immune cells from RRMS patients. While additional in vitro and ex vivo validation is needed, integrating patient transcriptomic data with structural biology already advances the field by prioritizing ITGB3 as a receptor of high interest. Similarly, our MD simulations, while informative, cannot capture the full conformational plasticity of integrins, which undergo large-scale transitions from bent to extended states upon ligand engagement [36]. These methodological constraints, however, do not undermine the conceptual advance: our study establishes ITGB3 as a credible molecular node linking peripheral immune dysregulation with neurovascular pathology in MS.

*ITGB3* sex-specific expression pattern may contribute to the clinical dimorphism of MS, while its therapy-dependent regulation could provide a readout of immunological adaptation under disease-modifying treatments. Targeting ITGB3 directly or modulating its interaction with PTN may open new avenues for precision therapies aimed at recalibrating immune–vascular crosstalk in MS. Beyond MS, these findings expand the scope of PTN biology, positioning pleiotrophin as not only a neurotrophic cytokine but also a systemic immunomodulator acting through integrin signaling.

## 4. Materials and Methods

### 4.1. Patients and Healthy Control Subjects

A total of 75 patients diagnosed with RRMS and 42 HCS were included. Clinical and demographic characteristics are summarized in Table 1. Patients participated in this analytical and transversal study conducted in the Neurology Department of the Western National Medical Center’s Specialty Hospital, Mexican Institute of Social Security (IMSS) in Jalisco, Mexico. Patients included in fulfilled the following criteria: (a) age between 20 and 60 years, (b) diagnosis of RRMS based on the 2018 revised McDonald criteria [37], (c) treatment with IFN-β, FINGO, NAT and RIT for at least three months or no treatment for at least two years. Clinical disability was assessed using Kurtzke’s Expanded Disability Status Scale (EDSS) [38]. The clinical form of RRMS was determined according to Lublin and Reingold classification [5,39].

Clinical relapse is defined by an acute deterioration in neurological function, while clinical remission means a period of partial or complete recovery without relapse episodes in the three months prior to enrollment in the study. The study does not include patients with the following conditions: (1) secondary progressive MS or primary progressive MS, (2) corticosteroid treatment in the last three months, (3) a history of autoimmune or inflammatory disease, and/or (4) another chronic degenerative CNS disease. All patients gave their written informed consent to participate in the study.

The control group consisted of 42 healthy individuals (females, n = 24 and males = 18), matched for age and sex, and collected from the IMSS Western National Medical Center’s Specialty Hospital Central Blood Bank. Blood samples from RRMS patients and HCS were taken at the same hour to minimize bias due to the circadian cycle. Control group samples were collected on different days but at the same hour. This study was conducted in accordance with the ethical guidelines of the 2024 Declaration of Helsinki and was approved by the Ethics and Investigation Committees of the IMSS (R-2022-785-045 and R-2025-785-049) and the Ethics, Investigation and Biosafety Committees of the CUCS (CI-01623) in Mexico. All participants gave their written informed consent to participate in the study.

### 4.2. Collection of Blood Samples and Isolation of Peripheral Blood Mononuclear Cells

Whole blood was collected in Vacutainer^®^ tubes (BD, New York City, NY, USA) containing ethylenediaminetetraacetic acid (EDTA) for total RNA extraction from isolated PBMCs. These cells were separated by density gradient centrifugation using Lymphoprep™ (STEMCELL Technologies, Vancouver, BC, Canada) according to the manufacturer’s protocol.

### 4.3. RNA Extraction and Evaluation of RNA Integrity by Agarose Gel Electrophoresis

Total RNA was isolated from PBMCs using the Trizol technique (Invitrogen, (Carlsbad, CA, USA, Ref. No. 15596018), followed by phase separation with chloroform (Sigma Aldrich, St. Louis, MO, USA, No. 288306), isopropanol precipitation (IBI Scientific, Dubuque, IA, USA, CAS# 67630), and washing with ethanol (IBI Scientific, CAS# 64175) as described by [40].

The concentration of total RNA was suspended in DEPC-treated, DNase/RNAse-free water (Invitrogen, No. AM9916) and analyzed by A260/280 absorbance using a NanoDropTM One Microvolume UV-Vis spectrophotometer (Thermo Fisher Scientific Inc., Waltham, MA, USA, catolog No. ND-ONE-W). RNA integrity was assessed by non-denaturing 2.5% agarose gel electrophoresis, 1X TAE buffer [41,42]. Samples (2 µL, 2 ng/uL) were loaded with DNA SYBR™ Safe (Invitrogen No. S33102), and the presence of 28S and 18S rRNA bands (4000 bp and 1600 bp, respectively) was used to confirm RNA quality (Appendix A). A 28S/18S ratio of ~2:1 indicates intact RNA. The RiboRuler High Range Marker (Thermo Fisher, SM1821) was used to confirm the size. Gels were visualized using an Analytik Jena UVP ChemStudio system, Jena, Germany.

### 4.4. cDNA Synthesis and Quantification of Receptors by RT-qPCR

Total RNA was reverse transcribed using the TaqMan™ Reverse Transcription Kit (Applied Biosystems, Foster City, CA, USA, N8080234) according to the manufacturer’s protocol. Reactions contained random hexamers, oligo(dT)_16_, and sequence-specific primers. The cDNA synthesis was quantified using a NanoDrop™ spectrophotometer (Wilmington, DE, USA). Sample concentrations were standardized to 140 ng/µL and 500 ng/µL to minimize bias in downstream reverse transcription quantitative polymerase chain reaction (RT-qPCR) analyzes.

Relative quantification was performed using 18S and GAPDH as endogenous controls. RT-qPCR was performed on a LightCycler^®^ 480 system (Roche, Basel, Switzerland) using TaqMan™ Fast Advanced Master Mix (Applied Biosystems, 4444556) and TaqMan™ probes for each target: Human *RPTPZ1* assay, Hs00960146_m1, No. 4351370, Human *ALK* assay, Hs00608281_m1, No. 4448490, Human *SDC3* assay, Hs1568665_m1, No. 4351370, Human *ITGB3* assay, Hs1001469_m1, No. 4448490, Human *GAPDH* assay, Hs99999905_m1, No. 4333764T, Human *18S* assay, Hs99999901_s1, No. 4333760F. Detection and quantification of PTN receptors was based on the presence of mRNA in PBMCs and confirmed by RT-qPCR with specific oligonucleotides that can be identified via the NCBI database or in silico predictions.

### 4.5. Molecular Docking Between ITGB3 Receptor and C-PTN

Molecular docking between the ITGB3 receptor with PDB identification number: 2VDR and C-PTN with PDB identification number: 2N6F was performed according to the systematic methodology described below. Protein structures were obtained from the RCSB Protein Data Bank (https://www.rcsb.org/, accessed on 15 January 2024) [43] and were prepared using UCSF Chimera 1.18, by removing water molecules and non-relevant chains, adding polar hydrogens, and assigning Gasteiger charges. Based on previous studies [44], the MIDAS (metal ion-dependent adhesion site) region was analyzed as it plays a role in the coordination of metal ions such as Mg^2+^ and Ca^2+^, which stabilize ligands such as fibrinogen.

As a reference, the RGD-containing fibrinogen peptide (AKQRGDV) [16] was energy minimized with Chem3D 4.0 using the MM2 force field [45]. The peptide used for docking was PTN123–132 (KKKKEGKKQE), selected based on [17], which identified this segment as structurally defined and biologically active. Peptide docking was performed as an exploratory step to identify potential ITGB3 binding hot spots, and the predicted poses were cross-checked against known integrin–ligand motifs. Docking was configured using AutoDock Tools 4.2.6 to define the grid box over the integrin binding site (Appendix A). AutoDock Vina was run from the command line with a configuration file containing the same grid parameters as AutoDock 4 and ligand/protein inputs. Peptide-docking results were analyzed in BIOVIA Discovery Studio Visualizer 2024, focusing on binding affinities and interaction profiles, including hydrogen bonds, hydrophobic contacts, and other non-covalent interactions between ITGB3 and C-PTN.

In addition, protein–protein docking between ITGB3 and PTN was performed using full-length HDOCK [46], a hybrid docking approach that integrates template-based modeling with free docking algorithms. The protein-protein docking results were analyzed, first with the best scoring, and then via PDBsum focusing on the proximity to the MIDAS region.

### 4.6. Molecular Dynamics

Molecular dynamics (MD) simulations were performed to explore the interaction dynamics and conformational fluctuations of the ITGB3–PTN complex. To better understand the source of the structural fluctuations, the RMSD was also calculated excluding the first 59 residues of ITGB3, which corresponds to a disordered N-terminal loop. This comparative analysis helped to determine whether the observed variability in the RMSD of the entire complex was caused by intrinsic flexibility in this region and not by general conformational changes. No separate MD simulation was performed with a truncated protein; the analysis was derived from the same trajectory.

MD simulations were performed using AMBER24 [47] to investigate the stability of the ITGB3–PTN complex, starting from the best scoring docking conformation modeled by HDOCK. The system was parameterized with the ff19SB force field and an OPC water model to evaluate the structural stability of the ITGB3–PTN complex, starting from the best-scored docking pose generated by HDOCK. The system was subjected to a two-stage energy minimization followed by heating and equilibration under NVT and NPT ensembles. Production simulations were limited for 250 ns in triplicate using the GPU-accelerated Particle Mesh Ewald Molecular Dynamics module (pmemd.cuda) and were intended as exploratory analyses of interaction capacity and fluctuation under simulated conditions, rather than assessments of full structural stability or convergence. Analysis of the trajectories included RMSD, hydrogen bonding, interatomic contacts, and MM-GBSA binding free energy estimates. To account for stochastic variation, three independent MD simulations were performed starting from the top-scoring docking poses. This approach allowed us to evaluate whether different initial conformations of the ITGB3–PTN complex could be stably accommodated under simulated conditions, providing exploratory insight into binding persistence rather than full convergence. The complete simulation parameters can be found in Appendix A.

### 4.7. Multiple Sequence Alignment and Conservation Analysis

Evolutionary conservation of the C-terminal region of PTN was assessed by multiple sequence alignment (MSA) using Jalview 2.11.4.0. Homologous sequences were determined via NCBI BLASTp v2.17.0, and those with high identity and coverage were selected. Alignment was performed using the Multiple Sequence Alignment Based on Fast Fourier Transform (MAFFT) algorithm implemented in Jalview and the BLOSUM62 substitution matrix. The analysis focused on the PTN123–132 region, which had previously been identified as the active docking peptide. Conserved residues were annotated and compared with interaction residues observed upon docking and in MD simulations to determine their structural and functional significance for the binding of ITGB3 in different species.

### 4.8. Statistical Analysis

A database was created using Microsoft Excel, and statistical analyzes were performed using GraphPad Prism version 10.0. Gene expression data from RT-qPCR were normalized using the 2^−ΔΔCt^ method [21]. After testing for normality, non-parametric statistical tests were performed due to the non-Gaussian distribution of the data. Comparisons between two groups were performed using the Mann–Whitney U test, while comparisons between more than two groups were performed using the Kruskal–Wallis test followed by Dunn’s Multiple Comparison Test. Correlations between non-normally distributed variables were assessed using Spearman’s rank correlation coefficient (rho). Statistical significance was set at *p* < 0.05.

## 5. Conclusions

In conclusion, our integrative study, which combines patient-derived PBMC expression data with molecular docking and dynamics analyses, identifies ITGB3 as a dynamic, sex- and therapy-modulated receptor of PTN in RRMS. This work challenges the CNS-centric view of PTN signaling and proposes a paradigm in which peripheral *ITGB3* expression reflects and potentially drives neuroinflammatory processes. By bridging clinical observations, immunological mechanisms, and structural modeling, we suggest that ITGB3 is a promising candidate for future mechanistic studies and therapeutic exploration in MS. The convergence of expression and structural data reframes pleiotrophin–ITGB3 signaling as a central axis of immune–neurovascular integration, underscoring its potential to reshape current strategies for diagnosis and treatment in multiple sclerosis.

## Figures and Tables

**Figure 1 ijms-26-12094-f001:**
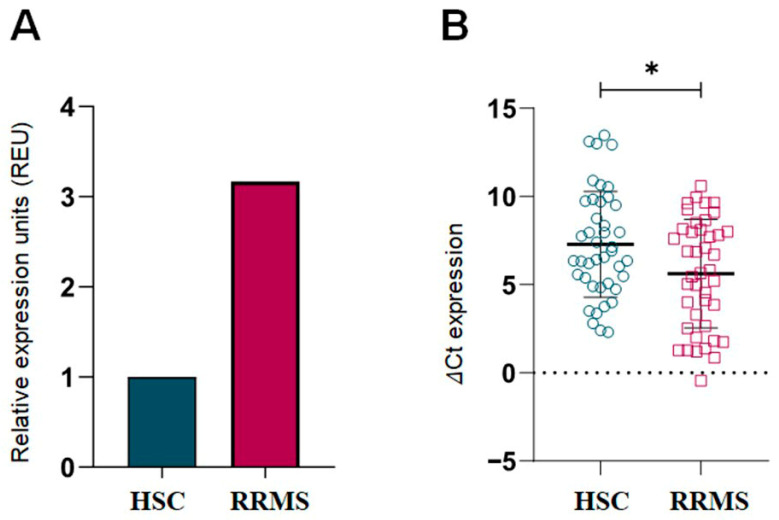
Relative Expression Units of *ITGB3* in RRMS patients and HCS. (**A**) Relative expression calculated by the (2^−ΔΔCt^) (Livak & Schmittgen, 2001 [21]) method, showing a 3.16-fold higher expression of *ITGB3* in the RRMS group (n = 42) compared to HCS (n = 42), normalized to the HCS group. (**B**) Individual ΔCt values of *ITGB3* expression in RRMS and HCS, showing the distribution of data and allowing statistical comparison. Data are expressed as mean (M) and standard deviation (SD). RRMS: Relapsing-Remitting Multiple Sclerosis; HCS: Healthy Control Subjects; * *p* < 0.05 (Mann–Whitney U test).

**Figure 2 ijms-26-12094-f002:**
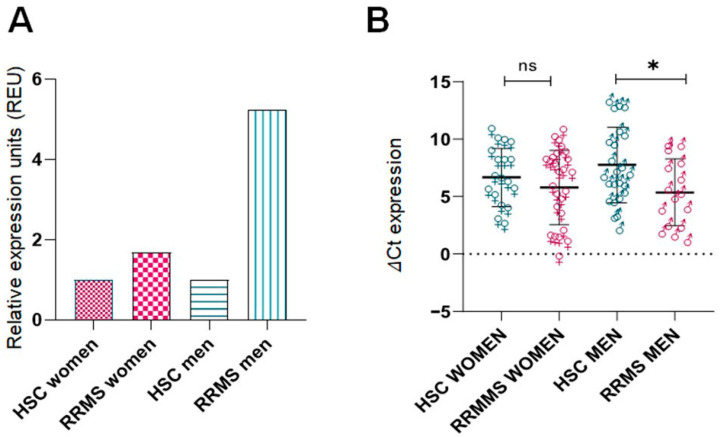
Sex specific differences in *ITGB3* expression in RRMS patients and healthy control subjects. (**A**) Relative expression of *ITGB3* (2^−ΔΔCt^) in four groups: RRMS females (n = 26), healthy females (n = 24), healthy males (n = 18), and RRMS males (n = 16). The expression values were normalized to their respective HCS groups (set to 1). Overexpression was observed in both disease groups: RRMS females showed a 1.68-fold increase compared to healthy females, and RRMS males showed a 5.2-fold increase compared to healthy males. (**B**) Comparison of *ITGB3* ΔCt values in the same four groups. The Kruskal–Wallis test showed no statistically significant differences between the groups overall. However, a comparison of the subgroups showed no significant difference between healthy and RRMS females (Mann–Whitney U test, *p* > 0.05), while a significant difference was observed between healthy and RRMS males (Mann–Whitney U test, * *p* < 0.05). RRMS: Relapsing-Remitting Multiple Sclerosis; HCS: Healthy Control Subjects.

**Figure 3 ijms-26-12094-f003:**
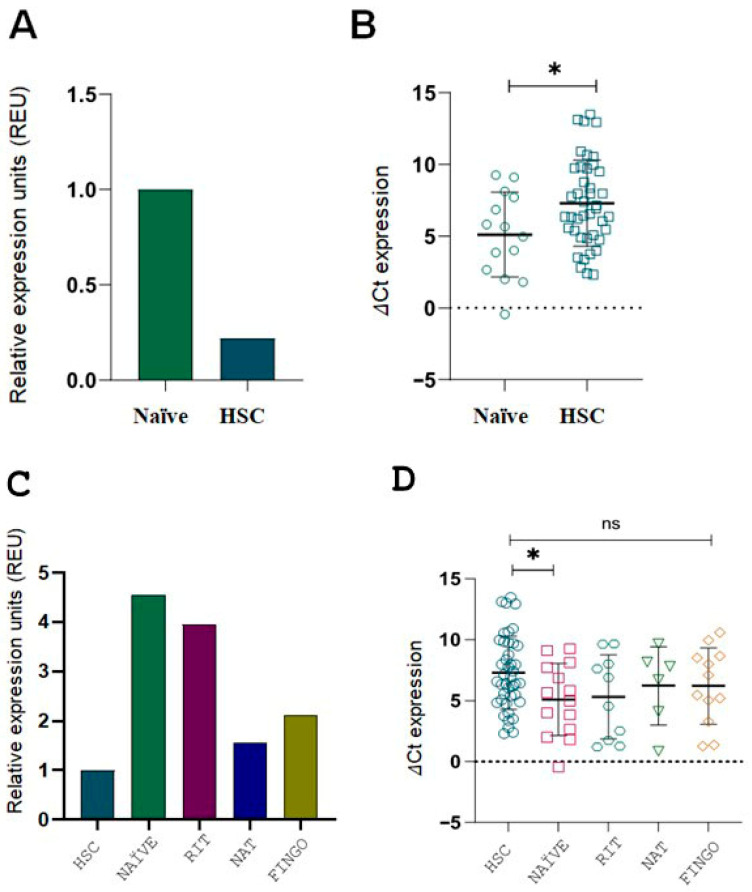
*ITGB3* expression in RRMS patients compared to HCS, including naïve and DMT-treated subgroups. (**A**) *ITGB3* expression measured by 2^−ΔΔCt^ is 0.22-fold lower in the HCS group (n = 42) than in the naïve group (n = 14). (**B**) ΔCt values for the same comparison confirm a significant difference (* *p* < 0.05, Mann–Whitney U test). (**C**) *ITGB3* expression levels (2^−ΔΔCt^ method) are shown relative to the HCS group, normalized to 1. Overexpression of *ITGB3* was observed in all patient groups: naïve (n = 14, 4.55-fold), RIT (n = 10, 3.94-fold), NAT (n = 6, 1.56-fold) and FINGO (n = 12, 2.12-fold), compared to the HCS group (n = 42, reference value = 1). Relative expression units (REU) are shown. (**D**) ΔCt values for the DMT and HCS groups show a significant difference only between HCS and the naïve subgroup (* *p* < 0.05, Mann–Whitney U test). Abbreviations: RRMS, relapsing–remitting multiple sclerosis; HCS, healthy control subjects; DMT, disease-modifying therapy; RIT, rituximab; NAT, natalizumab; FINGO, fingolimod; REU, relative expression units.

**Figure 4 ijms-26-12094-f004:**
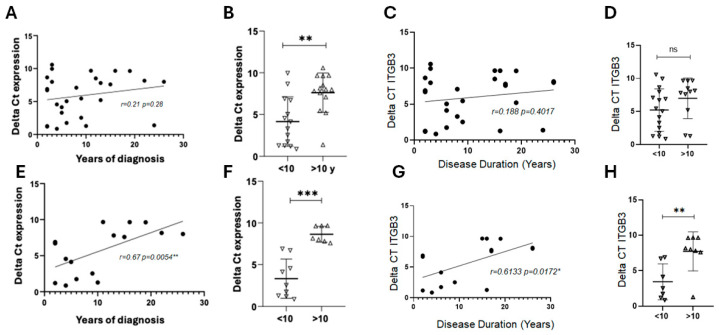
Correlation of *ITGB3* expression with years since diagnosis and disease duration in RRMS patients. (**A**) Correlation between *ITGB3* expression (ΔCt values) and years since diagnosis in the overall RRMS cohort under treatment (n = 28). Spearman’s correlation showed a weak positive trend (r = 0.21, *p* = 0.28). (**B**) Comparison of *ITGB3* expression between patients with less than 10 years (n = 14) and more than 10 years since diagnosis (n = 14). Mann–Whitney U test, ** *p* < 0.01. (**C**) Correlation of *ITGB3* expression with disease duration in the same cohort (n = 28). Spearman’s correlation showed no significant association (r = 0.188, *p* = 0.401). (**D**) Comparison of *ITGB3* expression between patients with <10 years (n = 16) and >10 years (n = 11) of disease duration. Mann–Whitney U test, *p* > 0.05, ns. (**E**) Correlation between *ITGB3* expression and years since diagnosis in patients treated with RIT (n = 10) and NAT (n = 6). Spearman’s correlation showed a strong positive association (r = 0.67, ** *p* = 0.0054). (**F**) Comparison of *ITGB3* expression in the same subgroup (<10 years, n = 9; >10 years, n = 7). Mann–Whitney U test, *** *p* < 0.001. (**G**) Correlation between *ITGB3* expression and disease duration in RIT- and NAT-treated patients. Spearman’s correlation showed a moderate positive association (r = 0.61, * *p* = 0.0172). (**H**) Comparison of *ITGB3* expression by disease duration (<10 years, n = 7; >10 years, n = 10). Mann–Whitney U test, ** *p* < 0.01. Abbreviations: RRMS, relapsing–remitting multiple sclerosis; ΔCt, delta cycle threshold; RIT, rituximab; NAT, natalizumab; ns, not significant.

**Figure 5 ijms-26-12094-f005:**
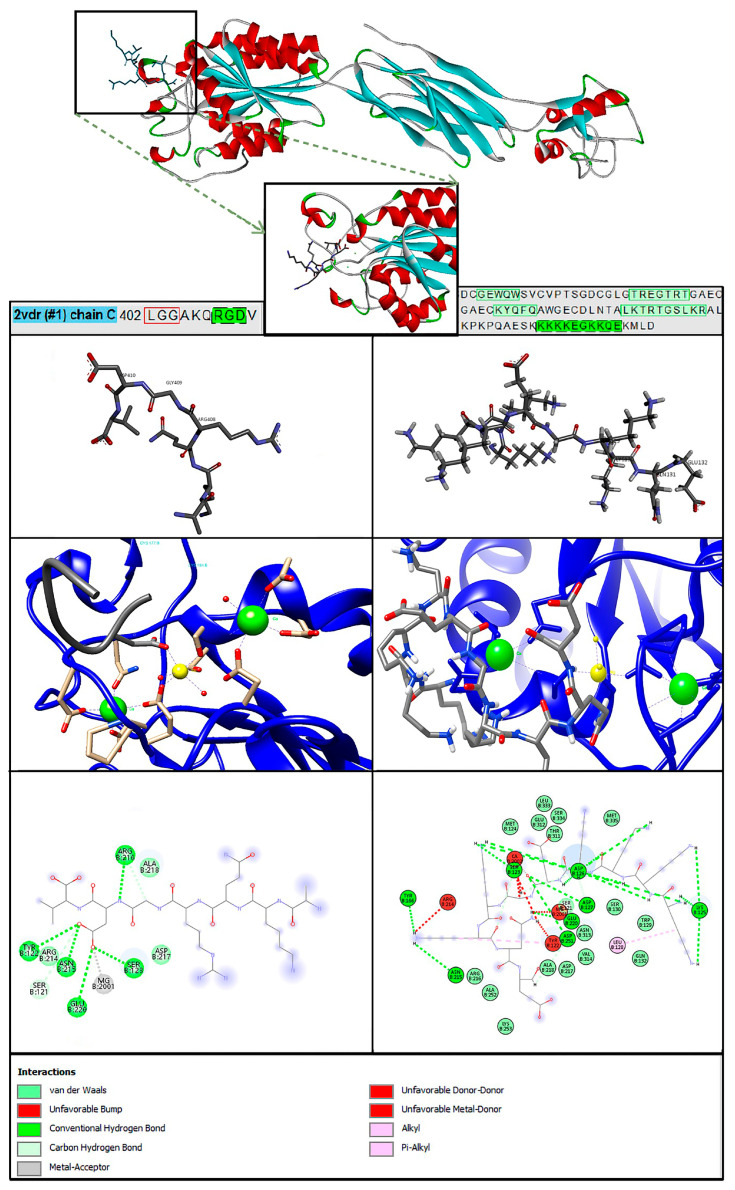
Molecular interactions of the ITGB3–PTN complex—docking results. The left column shows the images for fibrinogen binding, while the right column shows the images for PTN to validate the study with ITGB3. The ITGB3–fibrinogen complex (PDB 2VDR) was used to locate the biologically relevant binding site, guide PTN123–132 docking, and reference predicted interactions at the MIDAS site. Amino acid relevant sequences in green.

**Figure 6 ijms-26-12094-f006:**
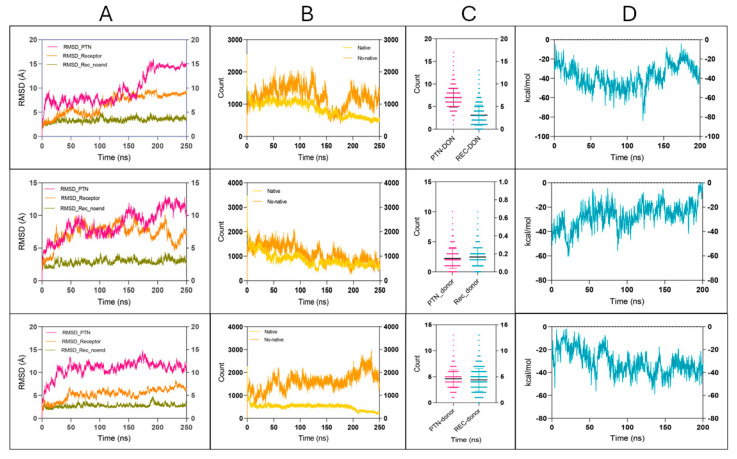
Molecular dynamics simulations of the ITGB3–PTN complex in three structural models, highlighting key parameters contributing to complex stability. Each row represents one of the three MD models (Models 1–3, from top to bottom), and each column corresponds to a specific analysis panel: (**A**) RMSD values for PTN (magenta), full-length ITGB3 (orange), and ITGB3 without the N-terminal region (residues 1–59, olive green), showing backbone stability over 250 ns. (**B**) Number of interatomic contacts (≤3.5 Å) between ITGB3 and PTN, sdistinguishing native contacts (light yellow) from non-native ones (dark orange). (**C**) Hydrogen bonds formed between PTN and ITGB3 over time, where PTN is the donor (pink) and ITGB3 is the acceptor (cyan). (**D**) Binding free energy (ΔG, kcal/mol) calculated using MM-GBSA over the entire 250 ns trajectory, showing persistent and favorable interaction energy profiles across all models.

**Figure 7 ijms-26-12094-f007:**
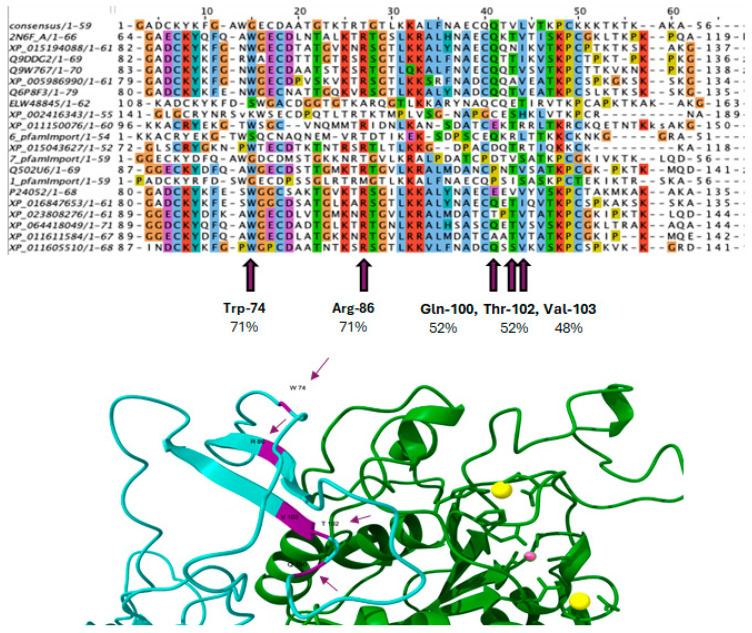
Mapping of conserved residues and structural interaction analysis of the PTN–ITGB3 complex. Top: MSA of the integrin-β3 binding region between species, highlighting conserved residues. Purple markers indicate residues involved in PTN binding based on docking and MD simulations. Bottom: Molecular dynamics simulation structure of PTN_112–136_ peptide (cyan ribbon) in complex with ITGB3 (green ribbon). Important conserved residues (Trp-74, Arg-86, Gln-100, Thr-102, Val-103) are shown in magenta as arrows, with percentages indicating sequence conservation. Yellow and pink spheres represent coordinated metal ions (Mg^2+^/Ca^2+^) that stabilize the interaction at the MIDAS site.

**Table 1 ijms-26-12094-t001:** RRMS: Relapsing-Remitting Multiple Sclerosis; HCS: Healthy Control Subjects; IFN-β: Interferon-beta; RIT: Rituximab; NAT: Natalizumab; FINGO: Fingolimod; EDSS: Expanded Disability Status Scale; BMI: Body Mass Index. Data are expressed in mean (M) ± standard Deviation (SD).

		RRMS	HCS
Total		75	42
Sex	Female	48 [64%]	24 [57%]
	Male	27 [36%]	18 [43%]
Treatment	Naïve	14	Not applicable
	IFN-β	33	
	RIT	10	
	NAT	6	
	FINGO	12	
EDSS 2.9		±1.9 (range 0–6)	Not applicable
Age (years)		43.1 ± 11.1	37 ± 9.6
BMI		25.5 ± 4.6	28.11 ± 3.5

**Table 2 ijms-26-12094-t002:** Summary of average values (mean ± SD) from the molecular dynamics simulations of the PTN–ITGB3 complex across three structural models. Parameters include RMSD for PTN and ITGB3 (full-length and truncated), number of interatomic contacts (classified as native—present in the initial docking pose—and non-native—newly formed during MD), hydrogen bonds (grouped by donor origin), and MM-GBSA binding free energy calculated over the full 250 ns trajectory.

Category	Label	Model 1 (Mean ± SD)	Model 2 (Mean ± SD)	Model 3 (Mean ± SD)
RMSD	RMSD_PTN	9.693 ± 3.215	8.653 ± 2.087	10.86 ± 1.679
RMSD	RMSD_Receptor	6.728 ± 2.096	7.220 ± 1.551	5.300± 1.199
RMSD	RMSD_Rec_noend	3.413 ± 0.4712	2.891 ± 0.4397	2.770 ± 0.359
Contacts	Native	855.3 ± 233.3	865.5 ± 272.0	502.2 ± 126.5
Contacts	Non-native	1251 ± 313.0	1116 ± 339.9	1605 ± 365.9
Hbonds	PTN_donor	6.976 ± 2.110	2.219 ± 1.638	4.614 ± 1.708
Hbonds	REC_donor	3.095 ± 2.225	2.472 ± 1.575	4.407 ± 2.182
MMGBSA	kcal/mol	−37.91 ± 12.18	−27.02 ± 10.33	−31.20 ± 10.74

## Data Availability

The original contributions presented in this study are included in the article/Appendix A. Further inquiries can be directed to the corresponding authors.

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
