# Peer review of "Overexpression of ITGB3 in Peripheral Blood Mononuclear Cells of Relapsing-Remitting Multiple Sclerosis Patients"

_ijms, 2025, doi:10.3390/ijms262412094_

Round 1
Reviewer 1 Report
Comments and Suggestions for Authors
Dear Authors,
The authors have demonstrated that the expression levels of PTN and its receptor ITGB3 were increased in male patients with RRMS.
Reviewer's comments as below.
(1) I'm curious why author focused on PTN even though this research group have shown as authors cited previous study (Reyes-Mata et al., 2021, Ref(14)). Please explain detail in this study if they done multiple screening to identify PTN as MS markers.
(2) Since the interaction between PTN and ITGB3 was investigated and published, structural analysis in Figure 5-7 are not helpful to establish author's story.
Author Response
Reviewer 1
Comment 1: I am curious why the authors focused on PTN, even though this research group has shown, as the authors cited in a previous study (Reyes-Mata et al., 2021, Ref (14)), that PTN is not associated with increased mortality risk. Please explain in detail in this study whether they performed multiple screenings to identify PTN as MS markers.
Response:
We thank the reviewer for this thoughtful comment. In our previous work (Reyes-Mata et al., 2021), we identified significant PTN overexpression in patients with multiple sclerosis, supporting the hypothesis that PTN participates in MS-related neuroinflammatory and neuroregenerative processes. That earlier study, however, was primarily descriptive and did not dissect the signaling pathways or receptor-mediated mechanisms through which PTN may contribute to disease biology.
The current manuscript represents the natural progression of that research, with the focused objective of characterizing the expression of the principal receptors mediating PTN signaling (PTPRZ, ALK, SDC3, and ITGB3) across distinct clinical phenotypes. Rather than conducting additional broad biomarker screens, we purposively selected PTN as a candidate given its established functional relevance in CNS homeostasis, oligodendrocyte maturation, and remyelination — pathways directly implicated in MS pathophysiology. We have revised the Introduction to explicitly articulate this rationale and contextualize the continuity between the two investigations.
Changes in manuscript
Lines 89-95: "Building on our previous findings that demonstrated PTN overexpression in RRMS patients relative to HCS [14], the present study aims to investigate the PTN signaling axis in greater mechanistic detail by examining the expression of its principal receptors (PTPRZ, ALK, SDC3, and ITGB3) in PBMCs. PTN was selected as a candidate molecule based on prior evidence of its relevance in neuroinflammatory and remyelination processes, and on the emerging role of microRNAs regulating PTN expression in MS [15]."
Comment 2:
Since the interaction between PTN and ITGB3 was investigated and published, the structural analysis in Figures 5–7 does not help establish the author's story.
Response:
We appreciate the reviewer's perspective. We agree that the PTN–ITGB3 interaction has been previously documented; however, the goal of our structural analyses was not to re-establish a known ligand–receptor binding event, but to provide a mechanistic context for our observed gene-expression signatures. Using molecular docking and dynamics simulations, we examined the energetic contribution, conformational stability, and key interface residues governing the PTN–ITGB3 interaction under biologically relevant conditions. This approach integrates structural modeling with patient-derived transcriptional data, strengthening the interpretation that increased ITGB3 expression may have functional consequences in MS.
We have revised the Discussion to more explicitly highlight how the structural data support the central narrative of the manuscript and contribute a novel dimension to the biological framework presented.
Changes in manuscript
Lines 397–405: "Although the PTN–ITGB3 interaction has been previously described, the objective of our analysis was to characterize the energetic and structural stability of the complex within the context of MS-related molecular alterations. By coupling molecular modeling with differential gene expression data, these findings provide an integrated perspective linking ITGB3 upregulation to PTN-mediated signaling. Docking studies revealed PTN binding at the MIDAS site with predicted affinities comparable to fibrinogen, the canonical endogenous ligand [28], thereby supporting the biological relevance of the PTN–ITGB3 interaction in disease-related settings."

Reviewer 2 Report
Comments and Suggestions for Authors
The manuscript explores ITGB3 overexpression in PBMCs of RRMS patients and integrates qPCR data with molecular docking and MD simulations to propose that PTN–ITGB3 interaction may be relevant in MS. However, several major issues weaken the manuscript.
Major:
1. The authors conclude that ITGB3 may mediate PTN-driven neuroinflammatory signaling, yet no experiments show that PTN actually signals through ITGB3 in PBMCs. Only mRNA expression and in-silico modeling are shown.
2. ITGB3 overexpression is shown only at the mRNA level.
3. Patients under multiple DMTs are included (IFN-β, RIT, NAT, fingolimod). IFN-β patients had zero detectable expression, a strong and surprising effect. This could confound the analysis and results. Please elaborate on this.
4. Authors show males have highest ITGB3 but do not analyze whether this is explained by treatment distribution or disease duration.
Minor:
1. Figure 1A and Figure 2A: The bar graphs should include individual data points and error bars to allow assessment of data distribution and variability. Additionally, Figure 1A does not specify the sample size (n) for the RRMS group in the panel or figure legend.
2. The abbreviation “Naïve” used in the patient subgroup figures may be misleading, since this refers to untreated RRMS patients, not healthy individuals.
3. The order of figure callouts is inconsistent. For example, Figure 4E is referenced before Figure 4C in the Results section.
4. In Figure 4, the rationale for selecting a cutoff of 10 years for subgrouping patients is unclear.
Author Response
Reviewer #2
Major Comments
Comment 1
The authors conclude that ITGB3 may mediate PTN-driven neuroinflammatory signalling, yet no experiments demonstrate that PTN signals through ITGB3 in PBMCs. Only mRNA expression and in-silico modelling are shown.
Response:
We appreciate the reviewer’s insightful observation. Our study indeed focuses on transcriptional characterization and structural modeling and does not directly demonstrate PTN–ITGB3 signaling in PBMCs. We have therefore revised the Abstract, Introduction, Results, Discussion, and Conclusion to explicitly position the study as exploratory and hypothesis-generating. We now clearly state that future work should include functional assays, such as ITGB3 blockade or knockdown during PTN stimulation, protein-level quantification, and receptor localization studies, to establish causality.
Changes in manuscript
- Abstract (lines 54–59): updated language to emphasize the hypothesis-generating nature.
- Introduction (end of final paragraph): added clarification of study scope.
- Discussion (limitations paragraph): updated to outline next steps and avoid causal language. Text modified throughout from causal to mechanistic hypothesis framing.
Changes in manuscript:
Abstract line 54-57, replace with: These findings suggest that ITGB3 is a dynamically regulated integrin receptor in RRMS that may participate in PTN-driven neuroinflammatory pathways in peripheral blood immune cells, influenced by disease stage, sex, and immunotherapy. While our results support the plausibility of PTN–ITGB3 engagement, they remain hypothesis-generating and require functional validation.
Introduction lines 115-118, add: Accordingly, the present study focuses on transcriptional profiling and in-silico structural analyses to prioritize ITGB3 within the PTN axis. We do not test receptor-mediated signaling experimentally; rather, we aim to generate mechanistic hypotheses to guide subsequent validation.
Discussion line 426-433, replace with: This study was conceived as an exploratory screening aimed at characterizing the gene expression profile of PTN receptors and providing structural validation through complementary bioinformatic analyses. Accordingly, ITGB3 expression was evaluated at the mRNA level, and molecular docking and molecular dynamics simulations were incorporated to strengthen the biological interpretation of the transcriptional data. While functional and protein-level validation were beyond the scope of this work, our integrated approach establishes a solid basis for subsequent studies investigating PTN–ITGB3 signaling mechanisms in immune cells from RRMS patients.
Comment 2
ITGB3 overexpression is shown only at the mRNA level.
Response:
We agree with the reviewer. Our results are constrained to gene-expression analyses, and protein-level validation was not included, due to the exploratory nature of this work. To support the biological plausibility of the transcriptional findings, we integrated molecular docking and dynamics simulations, thereby strengthening the interpretative framework that ITGB3 represents a compelling downstream candidate. This limitation is now explicitly acknowledged in the revised Discussion, linked to Comment 1 above, and highlights both the rationale and the planned future validation.
Comment 3
Patients under multiple DMTs are included (IFN-β, RIT, NAT, fingolimod). In IFN-β patients, there was zero detectable expression —a strong and surprising effect. This could confound the analysis and results. Please elaborate.
Response:
We thank the reviewer for raising this important point. We agree that the profound suppression of ITGB3 expression in patients treated with IFN-β warrants careful interpretation. This finding is consistent with well-established mechanisms by which IFN-β downregulates integrin-mediated leukocyte adhesion and migration. Accordingly, we have expanded the Results and Discussion to address treatment-related effects as a potential confounder and to emphasize that group comparisons should be interpreted within this pharmacological context. We have also cited supporting evidence (DOI: 10.1016/j.jneuroim.2004.01.002).
Changes in manuscript
- Results (lines 132-134): This suggest a treatment-related suppression consistent with the known immunomodula-tory effects of IFN-β, which downregulates integrin expression and reduces leukocyte ad-hesion and migration.
- Discussion (lines 385-387): This likely reflects the transcriptional downregulation of adhesion molecules induced by IFN-β, in contrast to other DMTs, such as NAT or FINGO, which act through distinct mechanisms and do not directly suppress gene expression.
These additions clarify that distinct DMT mechanisms may differentially modulate integrin expression, potentially including compensatory feedback upregulation under receptor blockade.
Comment 4
The authors report that males have the highest ITGB3 levels, but do not analyse whether this is explained by treatment distribution or disease duration.
Response:
We appreciate this methodological comment. Our statistical design included sex as a biological factor in all analyses, with additional correlation analyses performed for treatment type, disease duration, and diagnostic history. Given the exploratory nature and cohort size, independent analyses of each variable provided more interpretable biological insight than a multivariate model, which would risk overfitting in this context. Notably, the observed higher expression in males is consistent with established sex-related differences in MS immune activity and disease severity.
Minor Comments
Comment 1
Figures 1A and 2A should show individual data points and error bars; the sample size is missing.
Response:
We thank the reviewer for the suggestion. Because panels 1A–2A report fold change values normalized to the HCS mean (2^-ΔΔCt), individual dots would not represent statistically independent measurements and thus would not be meaningful. Instead, individual variability is appropriately depicted in panels 1B and 2B (ΔCt values), which served as the basis for statistical testing, consistent with qPCR reporting standards. Sample sizes (n = 42) have now been added to Figure 1A.
Changes in manuscript
- Line 156: included: (n=42) as sample size notation.
Comment 2
The abbreviation “Naïve” may be misleading.
Response – Preferred Option:
We appreciate this clarification. The term “treatment-naïve” is widely used in MS literature to indicate patients who have not received any DMT. To avoid ambiguity while maintaining terminological consistency with the field, we now explicitly define the term at its first occurrence. We support its use with representative citations (doi: 10.1212/WNL.0000000000209886; doi: 10.1016/j.msard.2025.106278; doi: 10.1177/1352458518790390).
Changes in manuscript
Definition added in its first occurrence as: naïve individuals (those who have not received any DMT). Consistent usage maintained.
Comment 3
Order of figure callouts is inconsistent.
Response:
We appreciate the reviewer’s attention to detail. We have ensured that all figure callouts follow numerical order in the revised text. Minor adjustments to panel lettering in Figure 4 ensure complete narrative-visual consistency.
Comment 4
The rationale for using a 10-year cutoff in Figure 4 is unclear.
Response:
We thank the reviewer for requesting clarification. The 10-year threshold was empirically guided by a distinct shift in the ΔCt distribution around this duration, suggesting an inflection point related to disease chronicity. This is also consistent with clinical evidence identifying approximately 10 years as a transition period from predominantly inflammatory to more neurodegenerative disease dynamics in RRMS. The rationale is now explicitly incorporated into the Results text and the Figure 4 legend.
Changes in manuscript
Lines 204-208: A justification for disease-duration stratification was added as follows: “Based on visual inspection of ΔCt data (Figure A,C,E,G), a noticeable shift in ITGB3 expression patterns was observed around 10 years of disease duration. This empirical separation used a 10-year cutoff to stratify patients, revealing significant differences in ITGB3 expression between early (<10 years) and long-standing (>10 years) RRMS.”

Round 2
Reviewer 2 Report
Comments and Suggestions for Authors
The authors have addressed my concerns and I have no more questions.
Author Response
Thank you for your thoughtful reviews and constructive comments, which will undoubtedly help us to improve the quality of our manuscript.